# SWE-Tester: Training Open-Source LLMs for Issue Reproduction in Real-World Repositories

## Abstract

Software testing is crucial for ensuring the correctness and reliability of software systems. Automated generation of issue reproduction tests from natural language issue descriptions enhances developer productivity by simplifying root cause analysis, promotes test-driven development – "test first, write code later", and can be used for improving the effectiveness of automated issue resolution systems like coding agents. Existing methods proposed for this task predominantly rely on closed-source LLMs, with limited exploration of open models. To address this, we propose **SWE-Tester** – a novel pipeline for training open-source LLMs to generate issue reproduction tests. First, we curate a high-quality training dataset of **41K** instances from **2.6K** open-source GitHub repositories and use it to train LLMs of varying sizes and families. The fine-tuned models achieve absolute improvements of up to **10%** in success rate and **21%** in change coverage on SWT-Bench Verified. Further analysis shows consistent improvements with increased inference-time compute, more data, and larger models. These results highlight the effectiveness of our framework for advancing open-source LLMs in this domain.

## 1 Introduction

Large language models have demonstrated remarkable improvements for software engineering (SWE) tasks which is evident from progress on benchmarks like SWE-Bench (Jimenez et al., 2024) which evaluates an LLM's ability to resolve issues in GitHub repositories. However, issue resolution is only one of the many responsibilities of a software developer which also includes writing tests, reviewing pull requests (PRs), developing system design etc. Although rigorous software testing is a critical component of the software development lifecycle, a large fraction of tests in repositories are added after issues are reported (Kang et al., 2023), and reproduction tests are rarely available in the test suite when an issue is reported (Kang et al., 2023; Mündler et al., 2024). Despite its importance, developers often dislike writing tests and consider it a mundane task (Straubinger & Fraser, 2023).

Given these observations, automating the generation of issue reproduction tests can significantly enhance developer productivity and facilitate test-driven development (TDD) which is known to improve code and test quality by encouraging tests to be written before source code modifications (Beck, 2022). Furthermore, reproduction tests are also leveraged by automated issue resolution approaches to rerank candidate bug fixes sampled from the LLM (Xia et al., 2024; Wei et al., 2025) and leverage inference-time compute using execution-based verifiers (Jain et al., 2025).

Prior work in this domain has proposed evaluation benchmarks like SWT-Bench (Mündler et al., 2024) and Defects4J (Just et al., 2014), and has developed various scaffolds for automated generation of issue reproduction tests (Ahmed et al., 2025a; Nashid et al., 2025; Kang et al., 2023; Wang et al., 2024; Khatib et al., 2025). Although effective, a crucial limitation of existing methods is their reliance on proprietary closed-source LLMs to achieve strong performance. While various efforts have been made to train LLMs for fixing issues by editing source code (Jain et al., 2025; Pan et al., 2025; Yang et al., 2025b; Xie et al., 2025), training LLMs for reproducing issues by writing reproduction tests is an unexplored research direction.

Furthermore, while issue resolution and issue reproduction are related tasks, both involving repository-level code generation, Mündler et al. (2024) observe that there is no correlation between

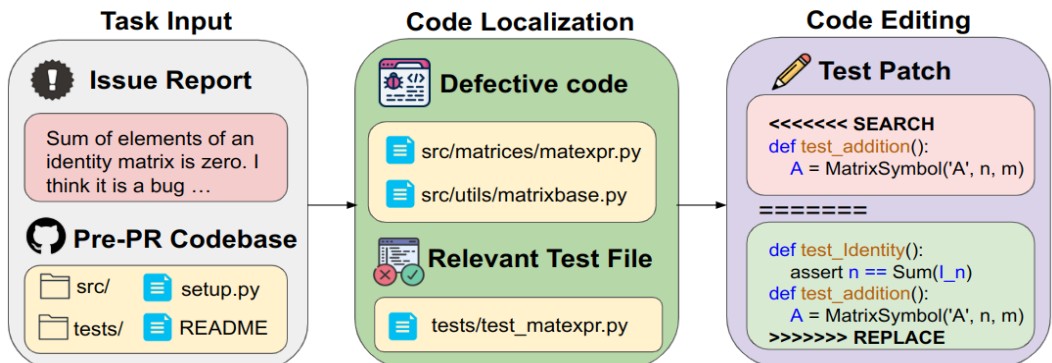

Figure 1: **Our proposed SWE-Tester pipeline.** Given an issue description and the pre-PR repository (which has this issue/bug), we first retrieve defective source code and relevant test file (code localization). Next, we edit the test file to augment it with reproduction tests (code editing).

an LLM's ability to fix an issue and reproduce it, and conclude that they are distinct tasks of different difficulty.

To address this research gap, we propose **SWE-Tester** – a novel framework for generating issue reproduction tests using fully open-source LLMs. As shown in Figure 1, we follow a standard two-step workflow consisting of code localization followed by code editing. Our analysis reveals that while open-source LLMs have reasonable code localization performance, they significantly struggle with code editing. Since code editing is the primary bottleneck in the workflow, we particularly focus on training LLMs to improve their code editing performance. We summarize our research contributions below:

1. **Training dataset for issue reproduction test generation**: We leverage raw PR data from open-source GitHub repositories to curate a **41K** training instances from **2.6K** repositories.

2. **Training LLMs using our dataset improves their ability to write comprehensive issue reproduction tests**: Our training recipe *generalizes* to five open-source LLMs of varying sizes and model families, Qwen-2.5 Coder Instruct (7B, 14B and 32B), Llama3.1-8B Instruct and Gemma3-12B Instruct, with absolute gains of upto **10%** in success rate and upto **21%** in change coverage on SWT-Bench Verified.

3. **Scaling inference-time compute and training data consistently improves model performance**: Our experiments reveal that model performance improves as we increase training data and inference-time compute and provide crucial insights for future improvements.

## 2 RELATED WORK

**Benchmarks and Training Methods for Issue Reproduction Test Generation.** Reproduction test generation is more challenging than unit test generation because instead of writing tests from existing code, tests must be derived from a natural language issue description and a buggy codebase, without access to the issue-fixing patch. Evaluation benchmarks for evaluating the ability of LLMs to write reproduction tests include SWT-Bench (Mündler et al., 2024) for Python repositories and Defects4J (Just et al., 2014) for Java codebases. While SWT-Bench uses *unit test* mode by default, where the LLM-generated tests must be integrated into repository's existing test suite, it also supports an easier *reproduction script* mode where the LLM generates a stand-alone Python script for issue reproduction. Prior training approaches are limited to the *reproduction script* mode. Jain et al. (2025) train LLMs as a testing agent using rejection sampling fine-tuning (Ahn et al., 2024) on trajectories sampled from expensive closed-source LLMs, while He et al. (2025) use reinforcement learning for a pipeline-based workflow, with limited performance gains due to unstable learning from sparse rewards. The *reproduction script* mode is unrealistic since developer-written PRs typically modify the existing test suite, making training data curation difficult for this mode. We train LLMs for *unit test* mode by leveraging developer-written tests in issue-resolving PRs.

| Category | Metric | Dataset |
|---|---|---|
| Size | # Instances | 41K |
| | # Repos | 2.6K |
| Issue Text | # words | 175.7 |
| Gold code patch | # Files edited | 1.4 |
| | # Lines edited | 37.2 |
| Gold test patch | # Files edited | 1 |
| | # Lines edited | 47.9 |

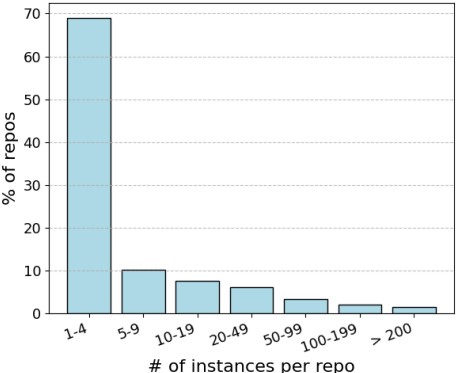

Table 1: Dataset statistics for our training dataset (left) and histogram of repo-level instance frequencies (right). Except for size, we report the instance-level averages for all metrics in the table.

**Workflows for Reproduction Test Generation** Prior approaches have proposed specialized pipeline-based workflows (Ahmed et al., 2025a; Kang et al., 2023; Khatib et al., 2025; Nashid et al., 2025) that often prompt a closed-source LLM for various steps like task planning, code localization, root cause analysis, iterative test refinement, etc., and have developed LLM agents like AEGIS (Wang et al., 2024) and SWE-Agent+ (Mündler et al., 2024). Furthermore, Mündler et al. (2024) also issue resolution systems like Aider (2024); Zhang et al. (2024) for this task. While effective, they often rely on closed-source LLMs for strong performance. Furthermore, these workflows are not suitable for training LLMs. For pipeline-based methods, there is no ground truth data available for most steps (e.g. planning, root cause analysis), which makes training data curation difficult. Similar problems arise when training LLMs for agent-based workflows since we need to sample trajectories from expensive closed-source LLMs to curate training data as done by Jain et al. (2025). We propose a simple, two-step pipeline-based workflow suitable for training open-source LLMs using raw PR data and achieve performance improvements for this task.

**Prior Work on Automated Issue Resolution.** Issue resolution is closely related to issue reproduction and involves modifying the source code to resolve issues given their natural language descriptions. Jimenez et al. (2024) develop SWE-Bench, a popular evaluation benchmark for this task, with multilingual (Yang et al., 2025b) and multimodal (Yang et al., 2025a) variants. Prior approaches for automating issue resolution propose LLM agents (Wang et al., 2025; Yang et al., 2024) and specialized pipelines (Xia et al., 2024; Örwall, 2024). Training methods for this task either train LLMs as coding agents (Pan et al., 2025; Jain et al., 2025; Yang et al., 2025b) or train LLMs for pipeline-based scaffolds (Wei et al., 2025; Xie et al., 2025; He et al., 2025). In this work, we train LLMs for reproduction test generation and curate training data by leveraging the raw GitHub PRs scraped by prior training methods for issue resolution like Xie et al. (2025); Pan et al. (2025) etc.

## 3 METHODOLOGY

In this section, we formally define the issue reproduction task in §3.1, describe our workflow in §3.2, explain our data curation and training methodology in §3.3, and finally describe the inference scaffold used to leverage inference-time compute in §3.4.

### 3.1 ISSUE REPRODUCTION TASK

We formally define the issue reproduction task following Mündler et al. (2024). Given a code repository $\mathcal{R}$, a natural language issue description $\mathcal{I}$, which is a bug report or a feature request in $\mathcal{R}$, and a golden code patch $\mathcal{X}^*$, containing source code changes from the issue-resolving PR. We denote the codebase $\mathcal{R}$ after applying a patch $\mathcal{X}$ as $\mathcal{R} \ o \ \mathcal{X}$. A test $s$ can either pass ($P$) or fail ($F$) when executed on $\mathcal{R}$. A test $s$ fails if its execution triggers an error (for eg: `AssertionError`, `ValueError`, etc.), and it passes otherwise. A test $s$ reproduces an issue $\mathcal{I}$ if it fails on the pre-PR codebase $\mathcal{R}$ and passes on the post-PR codebase $\mathcal{R} \ o \ \mathcal{X}^*$. Such tests are defined as fail-to-pass tests ($F \rightarrow P$), and $F \rightarrow F$, $P \rightarrow P$ and $P \rightarrow F$ tests are defined similarly. A set of tests $T$ reproduces an issue if: at least one test in $T$ is $F \rightarrow P$ (i.e it correctly exposes the issue), and no test in $T$ fails on

$\mathcal{R} \, o \, \mathcal{X}^*$ (avoiding $F \to F$ and $P \to F$ tests with incorrect testing logic or syntax issues). Also, this allows $P \to P$ regression tests in $T$. Given the pre-PR codebase $\mathcal{R}$ and the issue description $\mathcal{I}$, the task of the LLM is to generate a test set $T$ that reproduces the issue.

## 3.2 WORKFLOW FOR ISSUE REPRODUCTION

In this section, we describe our two-step workflow for generating issue reproduction tests (Figure 1) using **fully open-source LLMs**. The first step is code localization – retrieving relevant source code and test code from the pre-PR codebase. The second step is code editing – generating a patch to modify the retrieved test code and augmenting it with issue reproduction tests.

### 3.2.1 CODE LOCALIZATION

For code localization, we retrieve potentially defective source code files and a relevant test file from the pre-PR codebase. The source code files give context on possible root causes and highlight segments that require further testing for issue reproduction. The test file helps the LLM to understand existing testing conventions, identify gaps in testing logic critical for issue reproduction, and reuse relevant utility functions and classes.

Prior pipeline-based approaches follow similar localization steps, with some localizing to more fine-grained locations (e.g., classes or functions) by prompting an LLM (Khatib et al., 2025; Ahmed et al., 2025a). We provide the complete content of the retrieved files following training methods for issue resolution, which find that including the entire file contents improves performance by teaching the model to identify relevant code segments (Xie et al., 2025; Wei et al., 2025). For test code, we follow prior work (Ahmed et al., 2024; 2025a;b), that achieve strong performance on this task, and localize a single test file. While this may miss scenarios like utility code and tests being edited in separate files, it simplifies localization during inference and aids training data filtering (§3.3).

For source code, we adopt the coarse-to-fine retrieval strategy of Xie et al. (2025): BM25 (Robertson et al., 2009) retrieves the top-30 relevant source code files using issue description as the query, and a fine-tuned Qwen2.5-7B (Team, 2024) model then selects the defective files given skeletons of these 30 files. For test code, we use BM25 to retrieve the top-30 candidate test files with issue being the query, which are then reranked using Qwen3-Embedding-8B (Zhang et al., 2025) model. Since the top-ranked file is not always correct, we use an inference scaffold (§3.4) that samples multiple test patches from the LLM for the top-K test files and reranks them to select the most appropriate patch. Although we can train a retriever model for test files, we find code editing step to be the main bottleneck (§5.1).

### 3.2.2 CODE EDITING

The next step is code editing, where the LLM generates a patch to add reproduction tests to the test file given the issue and the localized context. A key design choice is the patch format: while the unified diff format (used in `git diff`) is a standard representation, it is highly sensitive to mis-specifications since LLMs often struggle with precise line numbers (Jimenez et al., 2024; Mündler et al., 2024). Mündler et al. (2024); Ahmed et al. (2025a) use LLM-friendly formats that only allow insertion and replacement of functions, but they do not support editing other code segments like imports, global constants, class attributes, etc. This is particularly problematic for training data curation because edits to the test suite in developer-written PRs are not restricted to function-level edits. To overcome these limitations, we use the Search/Replace format (Xia et al., 2024; Wei et al., 2025), which can represent any arbitrary file edits (example shown in Appendix A) and simplifies training data curation §3.3.

## 3.3 DATASET CURATION AND TRAINING METHODOLOGY

Next, we describe our data curation and training methodology for the code editing step. Prior training methods for issue resolution have scraped raw PR data from GitHub repositories, with some also developing sandboxed execution environments (Pan et al., 2025; Jain et al., 2025). While they target issue resolution, we instead train models to predict the test patch for issue reproduction – a related but distinct task, since an LLM's ability to fix an issue does not correlate with its ability to reproduce it (§1). During training, the LLM is prompted with the issue description, the pre-PR contents of the

gold defective source code and gold test files edited by the issue-resolving PR. Refer to Appendix A for the exact prompts used in our work.

To curate training data, we leverage PR data curated by SWE-Gym (Pan et al., 2025), SWE-Fixer (Xie et al., 2025), SWE-rebench (Badertdinov et al., 2025), and SWE-Bench Extra (Badertdinov et al., 2024), which contain a total of 140K issue–PR pairs across 5K repositories. However, these datasets do not provide pre-PR file contents of the source code and test files edited by the PR, so we scrape them by cloning the pre-PR codebase for each instance. Since raw PR data is very noisy, we exclude PRs that satisfy any of these conditions: 1) belongs to repositories present in SWT-Bench (our evaluation set), 2) edits non-Python files, 3) edits more than three or zero source code files, 4) does not edit *exactly* one test file, or 5) has empty issue descriptions. Since not every edit made by the test patch adds a reproduction test, filtering criterion 4) helps ensure that the edits are *generally* relevant to the issue. We further discuss our rationale behind filtering criterion in Appendix C. After filtering and de-duplication, we obtain **41K** high-quality training instances from **2.6K** repositories. We write custom scripts to translate developer-written test patches from the unified diff format to Search/Replace format. Table 1 shows dataset statistics and repo-level instance frequencies. Similar to our parent datasets, our data also has a long-tail distribution: while **70%** repositories have $<5$ instances, the most frequent repository (`pandas`) has 1529 instances.

## 3.4 INFERENCE SCAFFOLD

Our localization pipeline (§3.2.1) retrieves defective source files using SWE-Fixer 7B Retriever Xie et al. (2025) and reranks test files with Qwen3-Embedding-8B (Zhang et al., 2025). Since the top-ranked test file may not be correct, we generate $N = 8$ patches with temperature 0.5 for each of the top-$K = 4$ test files, yielding 32 candidates. All candidates use the same source code context from SWE-Fixer, as prior work (Mündler et al., 2024; Ahmed et al., 2025a) shows test localization has greater impact on model performance than source localization.

From these 32 candidates, we select the most appropriate patch through a multi-step filtering and reranking process. First, we discard all patches that are not applicable to the test file. Next, for each applicable patch, we execute the original test file and the modified test file (after applying the patch) on the pre-PR codebase. We compare the test execution logs of the two runs and discard all patches which introduce errors (e.g. `SyntaxError`, `ImportError`, etc.).

Since a reproduction test set must have at least one test that fails on the pre-PR codebase (§3.1), we only consider all such patches for the final patch set $\mathcal{S}$. We rerank patches in $\mathcal{S}$ using a self-consistency mechanism similar to (He et al., 2025). We assign a score to each patch $p$ in $\mathcal{S}$, computed as: $\text{Score}(p) = \text{Score}_{\text{EM}}(p) + \text{Score}_{\text{Sim}}(p)$. $\text{ScoreEM}(p)$ is the standard self-consistency measure, counting how often $p$ occurs in $\mathcal{S}$. $\text{ScoreSim}(p)$ rewards candidates that lie in dense clusters of plausible variants: we compute pairwise similarity between patches with Python's `difflib.SequenceMatcher`, identify the top-$r$ nearest neighbors for each candidate, and average their similarity scores. Following He et al. (2025), we set $r = |\mathcal{S}|/2$. The patch with the highest combined score is selected as the final answer. We describe potentially better reranking methods in Appendix E

## 4 EXPERIMENTAL SETUP

In this section, we describe our experimental setup for training and the evaluation metrics used to benchmark model performance.

Given the training dataset from §3.3, we perform supervised fine-tuning of LLMs across various sizes and model families. In particular, we train Qwen2.5-Coder Instruct models of sizes 7B, 14B, and 32B (Hui et al., 2024), Llama3.1-8B Instruct (Dubey et al., 2024), and Gemma3-12B Instruct (Team et al., 2025). We only consider instances with upto 16K tokens (tokenized using tokenizer of Qwen2.5-Coder 7B Instruct), resulting in 23K training instances. Additional training details are provided in Appendix D.

To evaluate model performance for issue reproduction task, we use the SWT-Bench Verified dataset (Mündler et al., 2024) consisting of 433 instances. The following metrics are used to compare model performance: applicability ($\mathcal{W}$), success rate ($\mathcal{S}$), and change coverage ($\Delta\mathcal{C}$). Applicability

Table 2: Comparing the applicability ($\mathcal{W}$), change coverage ($\Delta\mathcal{C}$) and success rate ($\mathcal{S}$) base and fine-tuned LLMs on SWT-Bench Verified (Mündler et al., 2024) using our inference scaffold (§3.4)

| Model | Applicability (%, ↑) | | | Change Coverage (%, ↑) | | | Success Rate (%, ↑) | | |
|---|---|---|---|---|---|---|---|---|---|
| | base | SFT | Δ | base | SFT | Δ | base | SFT | Δ |
| Llama3.1 8B Instruct | 18.01 | 84.76 | **+66.75** | 1.42 | 22.28 | +20.86 | 0.23 | 6.24 | +6.01 |
| Gemma3 12B Instruct | **85.91** | **90.99** | +5.08 | 5.95 | **24.12** | +18.17 | 0.92 | 8.08 | +7.16 |
| Qwen2.5 Coder 7B Instruct | 40.42 | 84.99 | +44.57 | 3.23 | 21.30 | +18.07 | 1.15 | 8.31 | +7.16 |
| Qwen2.5 Coder 14B Instruct | 30.95 | 84.99 | +54.04 | 2.51 | 23.54 | **+21.03** | 1.39 | 11.79 | **+10.40** |
| Qwen2.5 Coder 32B Instruct | 84.76 | 84.76 | 0.00 | **19.14** | 22.73 | +3.59 | **9.93** | **14.09** | +4.16 |

measures the percentage of instances where the LLM-generated patches are well-formed and apply cleanly to the codebase. While generating an applicable patch does not imply successful issue reproduction, it is a necessary pre-requisite for this task. Success rate measures the percentage of instances where the LLM-generated test set reproduces the issue (§3.1). Change coverage measures the fraction of executable lines in the golden code patch $\mathcal{X}^*$ that are executed more frequently in the modified test suite (after applying the LLM-generated patch) than in the original test suite. We refer the reader to (Mündler et al., 2024) for more details about these metrics.

## 5 RESULTS

Table 2 presents the experimental results of the proposed workflow and we describe the key takeaways below.

**Training LLMs using our dataset improves their ability to write issue reproduction tests:** Our proposed training recipe substantially enhances success rate and change coverage on SWT-Bench Verified. Notably, our approach **generalizes** to five open-source LLMs of varying sizes and model families, highlighting its effectiveness. While smaller base LLMs (with $\leq$ 14B parameters) have poor change coverage ($\leq 5.95\%$) and success rate ($\leq 1.39\%$), their fine-tuned variants achieve significant performance gains of upto **21.03** points in change coverage and **10.40** points in success rate. Furthermore, while the improvements over base LLM are comparatively lower, but still significant, for Qwen2.5-Coder 32B Instruct, we hypothesize that this is primarily due to limited training data (§6.3). Notably, these improvements are achieved using **fully open-source LLMs** without relying on proprietary models for any step in the workflow or for training data curation.

**Performance of fine-tuned LLMs scales with size, but model family also matters:** Rather unsurprisingly, fine-tuned models with more parameters generally have better success rate, with roughly similar change coverage across all models. Within the same family (Qwen2.5 Coder Instruct), the performance of the fine-tuned LLMs increases with size. Importantly, the choice of base model significantly impacts performance. Despite having comparable size, Llama3.1 8B Instruct has lower success rate (-2.07 points) than Qwen2.5 Coder 7B Instruct, and Gemma3 12B Instruct has lower success rate (3.71%) than Qwen2.5 Coder 14B Instruct.

**Training improves the ability of LLMs to generate well-formed patches:** Generating well-formed patches is a necessary condition for reproducing issues. Even when sampling multiple times for each test file, all the base LLMs (except for Qwen2.5-Coder 32B Instruct and Gemma3 12B Instruct) still struggle to generate applicable patches. On the other hand, the fine-tuned LLMs generate valid patches for nearly all the instances which fit in their context window.[1] Interestingly, Gemma generates applicable patches for **85.91%** of instances but can only reproduce **0.92%** issues which means that applicability is clearly not sufficient for issue reproduction.

While the focus of our work is to investigate how we can improve the performance of open-source LLMs for this task, we include comparisons with prior methods in Appendix F.

---

[1] For 37 instances, the SWE-Fixer's retriever model does not predict valid filenames. Among the remaining instances, the retrieved files do not fit in the LLM's context window for around 26 instances.

## 5.1 RETRIEVAL PERFORMANCE FOR CODE LOCALIZATION

In this section, we compare the localization performance of different retrieval methods for source code and test localization described in our workflow 3.2. Notably, these results validate that code editing step is the *primary* bottleneck in the workflow and small-sized open-source LLMs demonstrate much stronger performance for code localization performance as compared to code editing.

**Source Code Localization** Table 3 compares different retrieval strategies for source code localization (§3.2.1) on SWT-Bench Verified. SWE-Fixer Retriever LLM (Xie et al., 2025) achieves high precision and recall, highlighting the effectiveness of their training approach for code localization. While BM25 Top-4 and Top-30 achieve comparable or better recall, their precision is much lower precision SWE-Fixer.

Table 3: Comparison of source code localization performance on SWT-Bench Verified

| Method | Precision (%) | Recall (%) |
|---|---|---|
| BM25 Top-1 | 39.6 | 43.6 |
| BM25 Top-4 | 16.9 | 59.7 |
| BM25 Top-30 | 3.4 | **87.7** |
| SWE-Fixer Retrieval | **58.7** | 58.7 |

**Test Localization** Table 4 evaluates retrieval strategies for test localization on SWT-Bench Verified. Since we localize and edit a single test file per prompt, recall matters more than precision as our scaffold (§3.4) can potentially filter patches to irrelevant test files. We compare top-$K$ retrieval using BM25 versus BM25 + reranking with Qwen3-Embedding 0.6B and 8B models (Zhang et al., 2025). Reranking substantially improves recall, particularly for smaller values of $K$, achieving up to 73.1% recall. The 0.6B model performs nearly as well as the 8B variant, with much lower computational costs.

Table 4: Comparison of precision (Prec.) and recall (Rec.) of BM25 and BM25 Top-30 + reranking using Qwen3 Embedding for test localization (all metrics in %)

| Top-$K$ | BM25 | | BM25 Top-30 + Reranking | | | |
|---|---|---|---|---|---|---|
| | | | 0.6B | | 8B | |
| | Prec. | Rec. | Prec. | Rec. | Prec. | Rec. |
| 1 | 30.0 | 27.2 | 51.5 | 46.6 | 53.3 | 47.9 |
| 2 | 22.7 | 39.7 | 32.6 | 57.4 | 35.0 | 60.7 |
| 4 | 15.6 | 53.5 | 19.2 | 65.4 | 19.7 | 67.0 |
| 8 | 9.8 | 64.7 | 10.9 | 70.9 | 10.9 | 71.9 |
| 16 | 5.9 | 74.1 | 5.9 | 73.0 | 5.9 | 73.1 |

## 6 ANALYSIS

In this section, we perform concrete analysis to study which factors influence model performance and how it can be improved further. First, we study the impact of scaling inference-time compute on model performance in §6.1, compare edit-only performance of LLMs under a perfect retrieval assumption in §6.2, and study the impact of scaling training data on performance in §6.3

### 6.1 EFFECT OF SCALING INFERENCE-TIME COMPUTE ON MODEL PERFORMANCE

In this section, we study how model performance changes with increased inference-time compute. We keep the number of retrieved test files fixed (i.e $K = 4$) and vary the number of patches sampled for each test file ($N = \{1, 2, 4, 8\}$), resulting in a total of 4, 8, 16, and 32 candidate patches, and select the most appropriate patch using our scaffold (§3.4).

As shown in Figure 2, we observe consistent improvements in performance for the fine-tuned 7B, 14B and 32B Qwen2.5 Coder Instruct models. Notably, both success rate and change coverage improve steadily as more patches are sampled, and neither metric shows any signs of saturation for models of all 3 sizes. These trends suggest that the use of additional computation during inference can further improve model performance.

### 6.2 COMPARING EDIT-ONLY PERFORMANCE OF LLMS WITH ORACLE RETRIEVAL

Since we train LLMs primarily for code editing step, we follow SWE-Llama (Jimenez et al., 2024) and SWE-RL (Wei et al., 2025) in comparing edit-only performance under "oracle" localization. We provide pre-PR contents of gold source code files and gold test file edited in issue-resolving PR (selecting the file to which Fail-to-Pass tests were added if multiple exist). While unrealistic since developers must also localize relevant files to tackle an issue, this setup allows us to compare the code-editing abilities of LLMs isolated from localization errors. We use a simpler inference method

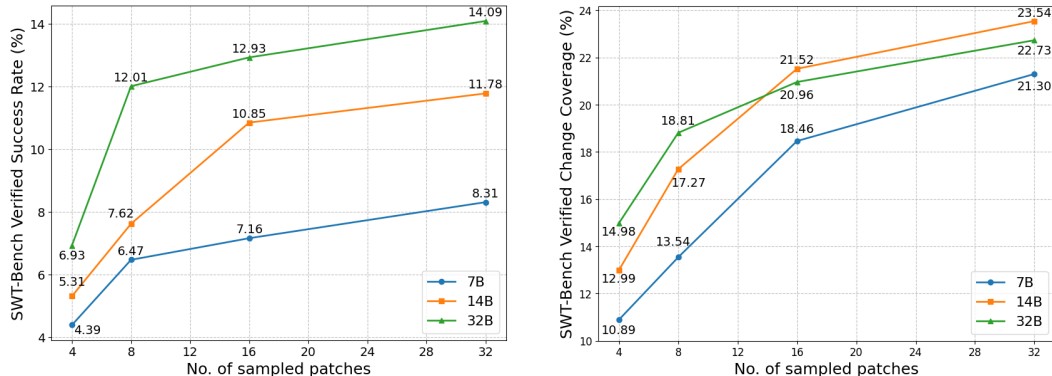

Figure 2: Increasing inference-time compute (i.e. no. of sampled patches) improves both success rate and change coverage of the fine-tuned Qwen2.5-Coder Instruct 7B, 14B, and 32B LLMs on SWT-Bench Verified.

for this setting compared to our scaffold §3.4. We keep sampling patches from the LLM until we get an applicable patch, starting at a temperature of 0, and gradually increasing temperature in steps of 0.1 for each retry, with maximum 10 retries. To minimize the effects of randomness, we report the average performance of 3 runs for base and fine-tuned Qwen2.5 Coder Instruct LLMs in Table 5. Here are the key takeaways:

**Base LLMs significantly struggle to reproduce issues even with oracle localization:** Base LLMs (particularly 7B and 14B variants) have significantly low applicability (despite 10 retries and perfect retrieval), low change coverage and low success rate making them practically unusable for this task. While the 32B base model achieves high applicability, it still underperforms its fine-tuned counterpart. Notably, the fine-tuned 14B model outperforms the base 32B model with **3.92%** higher success rate and **2.3x** fewer parameters.

**Sample efficiency of fine-tuned LLMs improves with better localization:** While it is expected that fine-tuned LLMs will perform better under an "oracle" retrieval assumption as compared to the setup in §5, an interesting insight is that they achieve this with **3.2x** fewer LLM samples than the original scaffold (§3.4). This suggests more sophisticated repository-level code localization approaches proposed by Chen et al. (2025); Sohrabizadeh et al. (2025); Wang et al. (2024) could improve sample efficiency for this task.

Finally, similar to our observations in §5, fine-tuned LLMs significantly outperform the base LLMs with improvements of upto **27.69%** in change coverage and **13.08%** in success rate. While the improvements observed for Qwen2.5 Coder 32B Instruct are relatively smaller, though still significant, we hypothesize this can be addressed by increasing training data (§6.3).

## 6.3 EFFECT OF SCALING TRAINING DATA ON MODEL PERFORMANCE

We analyze how editing performance scales with the amount of data used by our training recipe §4. We focus on 14B and 32B variants of Qwen2.5 Coder Instruct, the two best-performing models after training (§5, §6.2). We train both models by randomly sampling 5%, 25%, 50%, and 100% of the 23K training instances (§4), measuring success rate and change coverage on SWT-Bench Verified. Following Xie et al. (2025), we evaluate edit-only performance under oracle retrieval conditions (§6.2). The scaling trends are shown in Figure 3 and the key takeaways are described below:

**Increasing the amount of training data generally improves performance:** We observe that using more training data generally improves success rate and change coverage of 14B and 32B models. While, the performance of 14B model has plateaued after using 50% data, we observe no saturation signs in performance of the 32B model as we increase training data, especially for success rate. Therefore, using more training data is expected to improve performance of 32B model possibly due to higher learning capacity than its 14B counterpart.

Table 5: Comparing the applicability ($\mathcal{W}$), change coverage ($\Delta\mathcal{C}$) and success rate ($\mathcal{S}$) base and fine-tuned Qwen2.5 Coder Instruct LLMs on SWT-Bench Verified (Mündler et al., 2024) in an *edit-only* setup using "oracle" retrieval (§6.2)

| Model | Applicability (%, ↑) | | | Change Coverage (%, ↑) | | | Success Rate (%, ↑) | | |
|---|---|---|---|---|---|---|---|---|---|
| | base | SFT | Δ | base | SFT | Δ | base | SFT | Δ |
| 7B | $11.24_{\pm0.14}$ | $67.59_{\pm0.71}$ | **+56.35** | $2.28_{\pm0.16}$ | $26.42_{\pm0.71}$ | +24.14 | $0.46_{\pm0.00}$ | $9.24_{\pm0.80}$ | +8.78 |
| 14B | $16.73_{\pm1.31}$ | $\mathbf{70.75}_{\pm0.13}$ | +54.02 | $3.37_{\pm0.17}$ | $31.06_{\pm0.11}$ | **+27.69** | $1.93_{\pm0.27}$ | $15.01_{\pm0.61}$ | **+13.08** |
| 32B | $\mathbf{72.83}_{\pm0.13}$ | $68.90_{\pm0.75}$ | -3.93 | $23.26_{\pm0.07}$ | $\mathbf{31.15}_{\pm0.55}$ | +7.89 | $11.09_{\pm0.61}$ | $\mathbf{16.55}_{\pm0.70}$ | +5.46 |

**Significant performance gains are observed even with just 5% training data:** Interestingly, even when training with just 5% data, we observe significant improvements in success rate and change coverage over the base model. For the 14B model, we observe a significant absolute improvement of **8** points in success rate and **21.1** points in change coverage. For the 32B model there is an absolute improvement of **3** points in success rate and **4.5** points in change coverage.

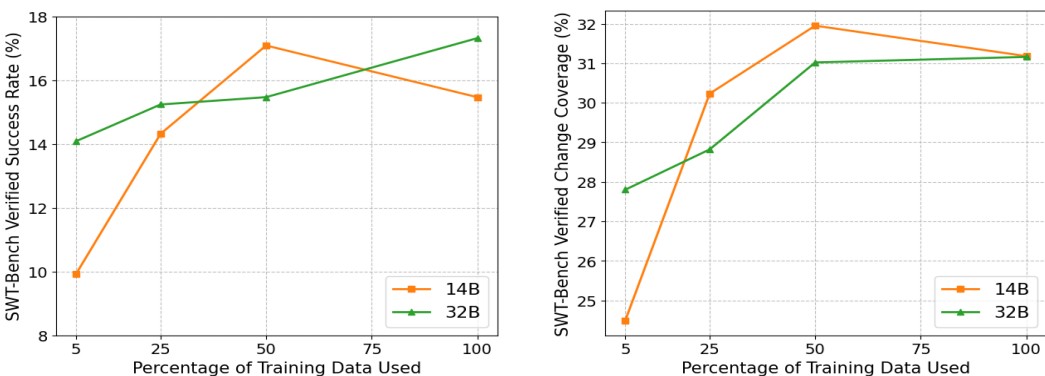

Figure 3: Scaling effects of increasing the amount data used when training Qwen2.5-Coder Instruct 14B and 32B LLMs.

### 6.4 ANALYSING PERFORMANCE ACROSS REPOSITORIES

We follow the setup from §5 and compute the percentage of issues reproduced across repositories for the fine-tuned Qwen-2.5 Coder Instruct models (7B, 14B, 32B). Table 8 shows interesting trends: the 7B model covers fewer repositories than 14B and 32B LLMs, with all models excelling on `scikit-learn` but struggling on `seaborn` and `sphinx`. Notably, 7B outperforms larger models on `pydata` and `xarray`, indicating that scaling does not always boost performance.

## 7 CONCLUSION

We propose **SWE-Tester** – a novel pipeline for leveraging open-source LLMs to generate issue reproduction tests. While many approaches have trained LLMs for issue resolution, training LLMs for issue reproduction is unexplored. We propose a two-step workflow suitable for training LLMs and curate a high quality training dataset of **41K** instances from **2.6K** repositories. Furthermore, our pipeline yields consistent performance gains on *five* open-source LLMs of varying sizes and model families demonstrating its generalizability. We perform comprehensive analysis to study the effects of scaling inference-time compute and training data, and analyze effect of retrieval quality on model performance, providing concrete insights for further improvements. While effective, our approach has its limitations (for eg., being limited to Python repos). We discuss our limitations and future work in Appendix J. In conclusion, our work provides a solid foundation for future research on training open-source LLMs for issue reproduction in real-world repositories.

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

## A  PROMPT TEMPLATE FOR ISSUE REPRODUCTION TEST GENERATION

We provide the prompt template used during training and inference below. During training, gold defective source code files and test code files are used. During inference, the files retrieved from the code localization step (§3.2.1) are used.

**Prompt template for generating issue reproduction tests**

**System Prompt**
You are a helpful AI assistant that can help reproduce a given GitHub issue by creating fail-to-pass test cases in the given test file. You MUST make necessary edits to the given test file to reproduce the issue. Do not try to fix the issue itself and do NOT change the source code. Your response format must follow the template below:

<solution>
Your final solution presented to the user as per user-defined *SEARCH/REPLACE* edit format.
</solution>

**User Prompt**
We are currently trying to reproduce the following Github issue in our repository. Here is the issue text:
<issue>
{Issue description}
</issue>

Below are some source code files that potentially contain bugs responsible for the issue.
<source code>
[start of source code file {filename1}]
{Contents of {filename1}}
[end of source code file {filename1}]

[start of source code file {filename2}]
{Contents of {filename2}}
[end of source code file {filename2}]
</source code>

Below is a relevant test file which you need to modify in order to reproduce the issue. Note that all your tests must be fail-to-pass tests i.e. they should fail on the given source code and should pass after the source code is fixed and the issue is resolved.
<test code>
[start of test code file {testfile}]
{Contents of the {testfile}}
[end of test code file {testfile}]
</test code>

**Prompt template for generating issue reproduction tests (Continued)**

**User Prompt (Continued)**
Present the test case(s) as a diff (SEARCH-REPLACE format, explained below). Each *SEARCH/REPLACE* edit must use this format:
1. The file path
2. The start of search block: <<<<<<< SEARCH
3. A contiguous chunk of lines to search for in the existing source code. The search string must EXACTLY match with the lines in the original source code (including indentation, comments, etc.) and there must be EXACTLY one unique occurrence of the search string in the source code.
4. The dividing line: =======
5. The lines to replace into the source code
6. The end of the replace block: >>>>>>> REPLACE
7. It must be encapsulated in a block ```python...```. Here is an example:
```python
### mathweb/flask/app.py
<<<<<<< SEARCH
from flask import Flask
=======
import math
from flask import Flask
>>>>>>> REPLACE
```

You can also have multiple *SEARCH/REPLACE* edits in the file where each edit is encapsulated in a separate ```python...```block. Please note that the *SEARCH/REPLACE* edit REQUIRES PROPER INDENTATION that is consistent with the existing code in the given test file. If you would like to add the line '     print(x)', you must fully write that out, with all those spaces before the code! The REPLACE string must be properly formatted with necessary indentation to ensure that the code is syntactically correct after the replacement.
IMPORTANT: Your final solution containing the *SEARCH/REPLACE* edits must be STRICTLY in the <solution> block and do not generate any text outside of the <solution> blocks.

## B  PROMPT TEMPLATE FOR RERANKING TEST FILES

We provide the prompt template for reranking test files using Qwen3 Embedding Zhang et al. (2025) below. Note that we use this instruction only for the query (issue) and the documents (test files) are embedded **without** this instruction.

**Prompt template for reranking test files retrieved using BM25 retrieval**

**Instruct**: Given a Github issue and a candidate test file, the given test file needs to be augmented with additional fail-to-pass test cases to reproduce the given issue i.e. the test cases need to fail on the current buggy codebase and pass after the bug is fixed. Determine if the given test file has relevant context (classes, functions, boilerplate code etc.) so that the downstream developer can easily augment the test file with minimal changes to reproduce the given issue.

**Query**: {Issue description}

## C    RATIONALE BEHIND FILTERING CRITERION FOR TRAINING DATA

In this section, we discuss our rationale behind the choice of filtering criterion during training data curation. While we can ideally use all the **140K** PRs for training, we have limited training compute which forces us to use stricter heuristics that may *potentially* discard possibly useful instances. We will release our source code for training data curation to enable future research on additional filtering methods. We list our reasoning for the five filtering criteria below:

1. **Filtering instances that belong to repositories in SWT-Bench dataset**: This is important to prevent any leakage and for analyzing if our method generalizes to repositories not seen during training. This criterion is used by all prior training methods for issue resolution (Pan et al., 2025; Jain et al., 2025; Yang et al., 2025b; Xie et al., 2025) etc.

2. **Filtering instances that edit non-Python files**: Firstly, this ensures that we discard trivial PRs that only edit documentation or README files. Furthermore, while Python is the main programming language used in the backend implementation, repositories may be using Javascript (for frontend), Dockerfiles for controlling sandbox settings, Bash scripts for utilities, etc. This criterion helps us ignore instances that edit portions of the source code other than the backend.

3. **Filtering instances which edit more than 3 or zero source code files** If the instances do not edit any source code files, the issues/PRs are probably low-quality/trivial since the core logic of the repository is unchanged. We follow SWE-FixerXie et al. (2025) and limit the number of edited source code files to 3 because more source code files. Firstly, SWE-Fixer contributes 110K PRs out of the total 140K PRs we scrape, and Xie et al. (2025) note that nearly 80% of their training data edits $\leq 3$ source code files. Furthermore, this helps us limit the proportion of context window used by source code file contents, and also potentially control the difficulty/complexity of the task because more source code files make it difficult to identify the root cause/bug and reproduce an issue.

4. **Filtering instances which do not edit exactly 1 test file** Firstly, instances that have edit zero test files are obviously not useful for our task. Furthermore, we observe that among the **87.6K** instances that edit at least 1 test file and at least 1 source code file, **66%** of instances edit only a single test file, implying that a large fraction of PRs have this property. Furthermore, Ahmed et al. (2025a; 2024) also include edit a single test file and achieve strong performance for this task. Therefore, editing a single test file is often sufficient to write reproduction tests. Since not every edit made to the test file is actually a reproduction test and there maybe regression tests or irrelevant code refactoring as well, restricting to PRs that edit a single test file ensures that they are *generally* relevant to the reported issue. Also this criterion helps us control the overall difficulty of the task considering base open-source LLMs have poor performance and are practically unusable for this task.

5. **Filtering instances with empty issue descriptions**: It is intuitively obvious that PRs from issues with empty descriptions will potentially add noise to our training dataset since there is no natural language description of the issue or the bug.

## D    ADDITIONAL TRAINING DETAILS

We train all the models for 2 epochs, with a batch size of 32, cosine learning rate scheduler, and AdamW optimizer (Loshchilov & Hutter, 2019). We set the learning rate to $1e^{-5}$ for Gemma3-12B Instruct and $3e^{-5}$ for all other LLMs. We use gradient clipping and set the maximum gradient norm at 1.0. We use torchtune library (torchtune maintainers & contributors, 2024) for all LLMs except for Gemma3-12B Instruct, where we use Llama-Factory library (Zheng et al., 2024) since torchtune does not have support for Gemma3 models. We use 2-8x A100 GPUs for training and inference.

## E    RERANKING STRATEGIES FOR INFERENCE SCAFFOLD

While self-consistency is widely considered for reranking patches by issue resolution systems (Xia et al., 2024; Wei et al., 2025; He et al., 2025), we hypothesize that better reranking approaches can be leverage for issue reproduction task.

Table 6: Comparison of success rates and change coverage (on SWT-Bench Verified) of SWE-Tester fine-tuned models with existing open-source, pipeline-based scaffolds that operate in unit-test mode. While our models have lower coverage than methods that use closed-source LLMs (potentially because we lose 63 instances due to limited context length), we have competitive success rates.

| Method | LLM | Success Rate (%) | Change Coverage (%) |
|---|---|---|---|
| **Closed-source LLMs** | | | |
| ZeroShotPlus | GPT-4o | 14.30% | 34% |
| LIBRO | GPT-4o | 17.80% | 38% |
| **Open-source LLMs** | | | |
| | Qwen2.5 Coder 7B Instruct | 8.31% | 21.30% |
| Ours | Qwen2.5 Coder 14B Instruct | 11.79% | 23.54% |
| | Qwen2.5 Coder 32B Instruct | 14.09% | 22.73% |

Our empirical results demonstrate that the change coverage of resolved instances is significantly higher (2-3 times more) as compared to that of unresolved instances, which indicates that change coverage offers a better scoring mechanism than self-consistency for reranking candidate test patches after filtering. However, this requires access to patches that edit the source code and attempt to resolve the issue since we do not have access to the gold source code patch from the issue-resolving PR. Ahmed et al. (2025b) use candidate source code patches from Agentless (Xia et al., 2024), a state-of-the-art issue resolution workflow, to compute change coverage of candidate test patches. However, since Agentless relies on closed-source LLMs and neither eOtter++ (Ahmed et al., 2025b) nor Otter (Ahmed et al., 2025a) release their source code, we leave better reranking methodologies for future work.

## F    COMPARISON WITH PRIOR APPROACHES FOR ISSUE REPRODUCTION

In this section, we include comparison with prior approaches on SWT-Bench Verified. In particular, we compare our approach with prior *pipeline-based* scaffolds for *unit test* mode that have open-source implementations in Table 6. While our models have lower coverage than methods that use closed-source LLMs (potentially because we cannot solve 63 instances due to limited context length and retrieval errors), we have competitive success rates. As observed in our empirical analysis, there is significant room for further improvement that can be achieved using various techniques like more inference-time compute, more training data and better inference scaffolds.

## G    CAN WE USE PEFT METHODS FOR TRAINING?

While most prior training methods generally train LLMs using full fine-tuning to train LLMs, we also investigate if we can achieve comparable performance by training models using parameter-efficient training methods, like LoRA (Hu et al., 2021). We train Qwen2.5 Coder Instruct models (7B and 32B variants) using LoRA (setting rank to 8 and alpha to 16, and follow the default settings in torchtune to train query, value and output projection layers). For other hyperparameters, training data etc., we use an identical setup as described in §4. Table 7 shows the results under an "oracle" retrieval setting. Notably, significant improvements are still achieved even with PeFT methods, demonstrating the feasibility of using our pipeline in limited computational budget.

## H    PERFORMANCE DISTRIBUTION ACROSS REPOSITORIES

## I    LICENSE OF DATA

The dataset used in this work is derived from publicly available GitHub pull requests collected by prior work. These repositories are distributed under permissive open-source licenses and our use of

Table 7: Results of training models using LoRA on applicability, change coverage, and success rate under base and SFT settings.

| Without Thinking | Applicability ( %, ↑) | | | Change Coverage ( %, ↑) | | | Success Rate ( %, ↑) | | |
|---|---|---|---|---|---|---|---|---|---|
| | base | SFT | Δ | base | SFT | Δ | base | SFT | Δ |
| Qwen2.5 Coder LoRA-7B | 10.62 | 69.28 | +58.66 | 2.03 | 25.74 | +23.71 | 0.69 | 10.39 | +9.70 |
| Qwen2.5 Coder LoRA-32B | 72.74 | 68.12 | −4.62 | 24.55 | 29.16 | +4.61 | 11.32 | 16.86 | +5.54 |

| Repository | 7B | 14B | 32B |
|---|---|---|---|
| astropy | 5.88% | 23.53% | 23.53% |
| django | 7.87% | 10.19% | 11.11% |
| matplotlib | 0.00% | 9.38% | 18.75% |
| seaborn | 0.00% | 0.00% | 0.00% |
| flask | 0.00% | 100.00% | 100.00% |
| requests | 25.00% | 0.00% | 0.00% |
| xarray | 26.67% | 6.67% | 6.67% |
| pylint | 0.00% | 16.67% | 33.33% |
| pytest | 13.33% | 13.33% | 26.67% |
| scikit-learn | 16.67% | 25.00% | 25.00% |
| sphinx | 3.57% | 3.57% | 3.57% |
| sympy | 8.22% | 13.70% | 16.44% |

Table 8: Model performance across repositories (percent of total issues reproduced) for fine-tuned Qwen2.5 Coder Instruct LLMs.

this data for research purposes is consistent with the terms of these licenses. In releasing our dataset, we will respect the original licensing conditions of the repositories from which the PRs were scraped. This ensures that downstream users can comply with the applicable open-source terms.

## J    LIMITATIONS AND FUTURE WORK

In this section, we discuss the limitations of our work. While our work provides a solid foundation for training open-source LLMs for issue reproduction, it has certain limitations, which we describe below.

First, our work is limited to Python repositories and a lot of prior methods have this limitation as well (Pan et al., 2025; Jain et al., 2025; Yang et al., 2025b). To the best of our knowledge, there exist open-source evaluation benchmarks for issue reproduction only for two programming languages – Defects4J (Just et al., 2014) for Java and SWT-Bench(Mündler et al., 2024) for Python which makes it difficult to evaluate training approaches for other programming languages. In addition, our work does not consider the impact of other filtering criterion for training data due to limited computational budget. However, we will release scripts for our dataset curation workflow that enables future work to analyse the impact of other filtering approaches. For test localization, we do not investigate how open-source LLMs can be trained for test localization since we achieve significant performance improvements even when using off-the-shelf open-source LLMs. We leave investigation on training

LLMs for test localization to future work. Furthermore, while our inference scaffold shows promising results, we do not investigate how we can further improve this scaffold through better reranking and patch filtering approaches, since the primary focus of our work is not to build the best systems, but to study how we can leverage/train open-source LLMs for issue reproduction. In addition, we cannot investigate if the developer-written test patches in training data actually reproduce the issue primarily because a large fraction of our dataset (88%) is obtained from SWE-Fixer (Xie et al., 2025) that does not create sandboxed execution environments with all dependencies configured as done by Pan et al. (2025); Yang et al. (2025b) etc. Note that creating execution environments is challenging since it either requires several hours of manual effort (Pan et al., 2025) or requires expensive calls to proprietary LLMs used by agent-based workflows that configure dependencies automatically (Yang et al., 2025b). Finally, while our assumption to a single test file simplifies localization and training, it is not ideal for scenarios where we require multi-file edits for issue reproduction, for eg. editing utility functions and tests in separate files. Future work can use our models and data to study if performance can be pushed further with reinforcement learning, reasoning-guided training (by leveraging chain-of-thought traces from stronger, proprietary LLMs), and train LLMs as effective testing agents.

## K   LLM USAGE

We used LLMs for polishing writing, improving clarity and for writing code to make some of the figures in this paper. All experiments, research ideas, analyses are conducted by authors.

