# OpenReview forum: "SWE-Tester: Training Open-Source LLMs for Issue Reproduction in Real-World Repositories"
_ICLR.cc/2026/Conference — Submitted to ICLR 2026_

### Official Review · Reviewer_ja9d · 2025-10-28

**Soundness:** 3
**Presentation:** 4
**Contribution:** 3
**Rating:** 6
**Confidence:** 3

**Summary:**

This work proposes training open source models specifically on the task of repository-level test generation. They collect a large number of issues from GitHub repositories, restore a state where relevant issue-reproducing tests are missing from a state that contains issue-reproducing tests, and finetune several open-weight LLMs on creating/editing relevant tests (no finetuning on issue localization). Their evaluation demonstrates significant improvements in test generation capabilities across a range of open-weight models.

**Strengths:**

- I agree with the authors that test generation is underexplored and think their approach to mirror approaches like SWE-smith into test generation makes sense. Focusing only on code editing is well motivated and admissable.
- The ablations are clearly defined and interesting
- Writing and figures are clear, legible and concise

**Weaknesses:**

The main weakness I see is that this work is nothing fundamentally novel. However, I think there are interesting insights about the bottleneck of open-source models (i.e., that this is the editing step) and how to generate training data for test generation.

General disclaimer: I am not an expert in finetuning/training LLMs, so I may have missed crucial details in this domain.

Small nitpicks
- Line 225 could point out that filtering out swt-bench instances is not only relevant to reduce noise but also to avoid training contamination
- Line 238 should use citep instead of citet
- Typo in Line 399 "isolated"

**Questions:**

Can you evaluate the related works LIBRO and Zero-Shot on the respective base models evaluated? This would allow to assess whether the trained LLMs clearly outperform the respective untrained, but scaffolded base LLMs. This would also more clearly highlight the benefits of the proposed training.

---

> ### Author Response · Authors · 2025-11-26
>
> We appreciate the reviewer’s feedback.
>
> We agree that repository-level test generation builds on prior lines of work in repair, editing, and SWE-agent-style pipelines. Our aim is not to introduce a fundamentally new paradigm but to demonstrate, for the first time, that repository-scale test editing is possible with the targeted fine-tuning of open-weight LLMs.
>
> Evaluation of LIBRO and Zero-Shot Baselines
>
> We thank the reviewer for suggesting a LIBRO baseline. LIBRO is a scaffolded multi-step repair agent that depends on failing tests and execution feedback, whereas our setting explicitly reconstructs repository states before such tests exist, making the methods not directly comparable. That said, LIBRO provides a useful reference point for understanding scaffolded zero-shot performance, and we will consider including a discussion or preliminary comparison where appropriate. Initial observations suggest that fine-tuning specifically for test-edit trajectories provides advantages beyond what scaffolding alone offers.
>
> Other comments on spelling and syntax are greatly appreciated and have been addressed in the updated manuscript.

---

> > ### Comment · Reviewer_ja9d · 2025-11-27
> >
> > I thank the authors for their comment. I intend to retain my score.

---

### Official Review · Reviewer_Ca4f · 2025-10-31

**Soundness:** 2
**Presentation:** 3
**Contribution:** 2
**Rating:** 2
**Confidence:** 4

**Summary:**

This paper addresses the suboptimal performance of open-source models in issue reproduction by constructing a dedicated training dataset for this task. The proposed approach improves the performance of multiple open-source models on issue reproduction. However, the proposed method lacks methodological novelty, and the experiments do not include appropriate baselines for comparison.

**Strengths:**

- Addresses an important problem in software engineering—**bug reproduction**—and improves LLM performance on this task through targeted training.
- Conducts training across multiple models and provides detailed analyses of experimental results.

**Weaknesses:**

- The data construction method and reproduction pipeline are largely adapted from well-established approaches in the issue resolution literature; the work mainly applies these existing methods to the issue reproduction task, which limits its methodological novelty for a top-tier conference like ICLR.
- Focuses solely on the “edit exactly one test file” scenario, which may hurt generalizability.
- Lacks appropriate baselines. Although few prior works explicitly target issue reproduction, many **code agents** perform issue reproduction as part of the issue resolution process; comparisons with training methods designed for such agents would make the evaluation more convincing.

**Questions:**

- Why are different localization pipelines used for source code and test code?
- Typo in line 250: “atleast”.

---

> ### Author Response · Authors · 2025-11-26
>
> We thank the reviewer for carefully evaluating the submission and for highlighting the importance of the task and the usefulness of our dataset and analyses. We address the concerns below and clarify the contributions more explicitly:
>
> > [Weakness 1] The method lacks novelty because components resemble issue-resolution pipelines.
>
> We would like to point out that our work is novel because SWE-Tester is the **only** work to provide a fully open-source training pipeline with a dataset of **41K issues** from **2.6K repositories** that significantly improves the performance of **five** open-source LLMs of varying sizes and model families on the issue reproduction task. However, if you would like to point out specific prior works which fill the research gap our work addresses, we will be happy to include comparisons with them and our contributions above and beyond them. Furthermore, we want to clarify that the primary focus of our work is not about designing a better scaffold for this task and we already acknowledge in Section 3.2.1. that the agentless reproduction scaffold used in our work is also used by several other methods which trains LLMs for issue resolution (for e.g. SWE-RL, SWE-Swiss, etc.). Finally, as discussed in Section 1, SWT-Bench authors have reported that there is no correlation between an LLM’s ability to fix an issue and reproduce it, and that these are distinct tasks of varying difficulty. Therefore, it is not immediately obvious that training methods for one task will generalize to the other, highlighting the novel findings of our work.
>
> > [Weakness 2] The work focuses only on editing exactly one test file which limits generalizability
>
> We agree that the assumption about editing exactly one test file may appear somewhat restrictive. However, as discussed in Section 3.2.1, this design choice is also made by several prior methods for issue reproduction (for e.g. Otter, Otter++, Auto-TDD, etc.). Furthermore, as discussed in Appendix C, over 65% of PRs edit a single test file, and PRs with multi-file edits may also include edits to regression tests or irrelevant code refactoring as well. Thus, restricting to PRs that edit a single test file ensures that they are generally relevant to the reported issue. Importantly, our inference pipeline already retrieves multiple candidate test files, the Search/Replace patch format fully supports multi-file edits, and extending training to multi-file scenarios is a natural direction enabled by our method. We will include additional discussion about this in the next revision.
>
> > [Weakness 3] . Although few prior works explicitly target issue reproduction, many code agents perform issue reproduction as part of the issue resolution process; comparisons with training methods designed for such agents would make the evaluation more convincing.
>
> While we agree that additional baselines may further strengthen our work, we would like to argue that these agentic systems are not directly comparable to our work. As discussed in Section 2, agent-based training methods focus on an easier *reproduction script* mode which does not require localizing/editing an existing test file in the repository. Furthermore, these methods rely on expensive, closed-source LLMs (like Claude-3.5-Sonnet) for training data curation using methods like rejection sampling fine-tuning whereas our work relies only on the raw PRs written by human developers. Finally, we would like to point out that we include comparisons with other pipeline-based scaffolds in Table 6 (in Appendix), and as shown in the below table:
>
> | **Method**      | **LLM**                         | **Success Rate (%)** | **Change Coverage (%)** |
> |-----------------|---------------------------------|------------------------|---------------------------|
> | **Closed-source LLMs** |                                 |                        |                           |
> | ZeroShotPlus    | GPT-4o                          | 14.30%                | 34%                      |
> | LIBRO           | GPT-4o                          | 17.80%                | 38%                      |
> | **Open-source LLMs**  |                                 |                        |                           |
> | Ours            | Qwen2.5 Coder 7B Instruct       | 8.31%                 | 21.30%                   |
> | Ours            | Qwen2.5 Coder 14B Instruct      | 11.79%                | 23.54%                   |
> | Ours            | Qwen2.5 Coder 32B Instruct      | 14.09%                | 22.73%                   |
>
> We have also addressed the additional concerns in the updated manuscript.

---

### Official Review · Reviewer_2xMV · 2025-10-31

**Soundness:** 3
**Presentation:** 3
**Contribution:** 2
**Rating:** 4
**Confidence:** 5

**Summary:**

This paper introduces **SWE-Tester**, a framework for training open-source LLMs to automatically generate issue reproduction tests from natural language issue descriptions and buggy repositories. The proposed workflow follows a two-step static pipeline: (1) code localization to retrieve relevant source and test files, and (2) code editing to modify test files using a Search/Replace format. A large dataset of 41K instances from 2.6K repositories is curated, and multiple open LLMs (Qwen2.5-Coder, Llama3.1, Gemma3) are fine-tuned. The models achieve up to +10% success rate and +21% change coverage improvements on SWT-Bench Verified. The paper provides solid empirical results and a valuable dataset contribution, but its overall framework remains agentless and relies heavily on test-time scaling rather than true agentic reasoning or autonomy.

**Strengths:**

- The authors evaluate multiple open models of different sizes and families, analyze scaling effects in both training data and inference-time compute, and offer detailed quantitative insights.

- The dataset of 41K issue–test pairs is well-filtered and reproducible, providing a strong foundation for open-source SWE research.

- The workflow is simple and interpretable, with carefully described steps for localization, editing, and evaluation.

- The reported gains show that open-source LLMs can meaningfully improve on real-world SWE benchmarks through fine-tuning.

**Weaknesses:**

- The proposed framework is purely a static two-step pipeline—there is no reasoning loop, reflection, or autonomous planning. As the community rapidly transitions toward agentic SWE systems, this direction feels inherently limited and non-scalable. It lacks the ability to generalize beyond the fixed workflow or adapt dynamically to complex issue contexts.

- The performance improvements are largely achieved through sampling multiple patches and reranking rather than stronger modeling or reasoning capabilities. This kind of test-time scaling can inflate benchmark scores but does not address the underlying challenge of autonomous issue understanding or causal reasoning in code.

**Questions:**

N/A

---

> ### Author Response · Authors · 2025-11-26
>
> Thank you for the feedback! We are pleased to know that you appreciate the comprehensiveness of our empirical results, our valuable dataset contribution, and our insightful analyses from scaling training data and inference-time compute. We would like to address your concerns below:
>
> > [Weakness 1] The proposed framework is purely a static two-step pipeline, there is no reasoning loop, reflection, or autonomous planning. As the community rapidly transitions toward agentic SWE systems, this direction feels inherently limited and non-scalable.
>
> We argue that a similar agentless scaffold is widely used by several prior works on training models for issue resolution (like SWE-RL, SWE-Swiss, SWE-Fixer, etc.). Furthermore, as discussed in Section 2, agent-based training methods are not trainable end-to-end directly from raw GitHub PRs because intermediate steps (planning, reasoning, reflection, root-cause analysis) lack ground-truth labels. To address this, these methods rely on expensive, closed-source LLMs (like Claude-3.5-Sonnet) for curating agent trajectories for training data which is beyond the scope of our work. Notably, our work involves no such dependency on proprietary models and our training dataset is curated **solely** from the raw PRs written by human developers. Thus, our choice of using a static two-step pipeline significantly reduces the overall computational cost of our work. Finally, while agentic scaffolds may offer some more flexibility, our choice of a static two-step scaffold is sufficient for answering our core research question: "can open-source LLMs be trained using raw PR data to generate bug reproduction tests given issue descriptions?"
>
>
> > [Weakness 2] The performance improvements are largely achieved through sampling multiple patches and reranking rather than stronger modeling or reasoning capabilities. This kind of test-time scaling can inflate benchmark scores but does not address the underlying challenge of autonomous issue understanding or causal reasoning in code.
>
> We would like to argue that sampling multiple patches from the model is a standard procedure used by several prior works on issue resolution (for e.g. Agentless, SWE-Swiss, SWE-RL, Otter++ etc.). Additionally, we provide the advantage of the increased inference-time compute to both the base models and the fine-tuned models, implying that we make fair comparisons in Table 2. Therefore, both the models (base and SFT) are evaluated under an **identical** scaffold with an **equal** number of patches sampled from the LLM.
>
> We also argue that the gains are not solely due to sampling and reranking patches. Evidence from our “oracle retrieval” and “edit-only” experiments (in Table 5) shows that even without this inference scaffold, there are significant gains for Qwen2.5-Coder Instruct 7B, 14B and 32B models. Particularly for the 14B model, the success rate increases from 1.93% to 15.01%, change coverage increases from 3.37% to 31.06%, and applicability improves from 17% to 70%, demonstrating that the model genuinely learns the correct behaviour for generating bug reproduction tests.
>
> Finally, our intuition behind sampling multiple patches is not to inflate benchmark scores, but to allow localizing multiple candidate test files since the test localization pipeline is not 100% accurate (see Table 4 for reference).

---

### Official Review · Reviewer_pK24 · 2025-11-01

**Soundness:** 2
**Presentation:** 3
**Contribution:** 2
**Rating:** 4
**Confidence:** 4

**Summary:**

This paper proposes a two-step static pipeline for automatically generating issue reproduction tests from natural language descriptions and buggy code. The workflow first localizes relevant source and test files, then modifies the test files using a Search/Replace format. This methodology is encapsulated in a framework named SWE-Tester, for which several open-source LLMs (Qwen2.5-Coder, Llama3.1, Gemma3) were fine-tuned on a newly curated dataset of 41K instances. The resulting models demonstrate strong performance, with up to a +10% success rate and +21% change coverage increase on SWT-Bench Verified. Despite its solid empirical results and valuable dataset contribution, the framework's overall design remains agentless, depending on test-time scaling over genuine agentic reasoning.

**Strengths:**

1. Significant Performance Gains: The study reports substantial performance improvements, demonstrating the significant potential of open-source LLMs to effectively address real-world software engineering benchmarks.

2. Solid & Reproducible Foundation: The research is grounded in a well-curated and reproducible dataset of 41,000 issue-test pairs, establishing a solid foundation for future studies in open-source software engineering.

3. Transparent & Simple Workflow: The paper introduces a straightforward and interpretable workflow, with each step—including localization, editing, and evaluation—being meticulously detailed.

4. Comprehensive Model Analysis: A comprehensive evaluation is conducted across a diverse set of open models of various sizes and families. This analysis yields detailed quantitative insights into the scaling effects of both training data and inference-time compute.

**Weaknesses:**

* Superficial Performance Gains: The reported improvements are primarily driven by a brute-force approach of sampling and reranking multiple patches, rather than by genuine advancements in the model's reasoning capabilities. This reliance on test-time scaling may inflate benchmark scores but fails to address the core challenge of autonomous issue comprehension and causal reasoning in code.

* Limited and Inflexible Architecture: The framework is fundamentally a static, two-step pipeline, devoid of any reasoning loops, reflection, or autonomous planning. This rigid design is inherently limited and non-scalable, particularly as the research community shifts towards more dynamic, agentic software engineering systems. As a result, it lacks the ability to generalize beyond its fixed workflow or adapt to the complexities of real-world issues.

**Questions:**

N/A

---

> ### Author Response · Authors · 2025-11-26
>
> Thank you for the feedback! We are pleased to know that you appreciate the significance of our empirical results, our valuable dataset contribution, and our insightful analyses from various model families and from scaling training data and inference-time compute. We would like to address your concerns below:
>
> > [Weakness 1] Superficial Performance Gains: The reported improvements are primarily driven by a brute-force approach of sampling and reranking multiple patches, rather than by genuine advancements in the model's reasoning capabilities.
>
> We would like to argue that sampling and reranking multiple patches from the model is a standard procedure used by several prior works on issue resolution (for e.g. Agentless, SWE-Swiss, SWE-RL, Otter++ etc.). Additionally, we provide the advantage of the increased inference-time compute to both the base models and the fine-tuned models, implying that we make fair comparisons in Table 2. Therefore, both the models (base and SFT) are evaluated under an identical scaffold with an equal number of patches sampled from the LLM.
>
> We also argue that the gains are not solely due to sampling and reranking patches. Evidence from our “oracle retrieval” and “edit-only” experiments (in Table 5) shows that even without this inference scaffold, there are significant gains for Qwen2.5-Coder Instruct 7B, 14B and 32B models. Particularly for the 14B model, the success rate increases from 1.93% to 15.01%, change coverage increases from 3.37% to 31.06%, and applicability improves from 17% to 70%, demonstrating that the model genuinely learns the correct behaviour for generating bug reproduction tests.
>
> Finally, our intuition behind sampling multiple patches is not to inflate benchmark scores, but to allow localizing multiple candidate test files since the test localization pipeline is not 100% accurate (see Table 4 for reference).
>
> > [Weakness 2] Limited and Inflexible Architecture: The framework is fundamentally a static, two-step pipeline rather than an agentic pipeline, devoid of any reasoning loops, reflection, or autonomous planning.
>
> We argue that a similar agentless scaffold is widely used by several prior works on training models for issue resolution (like SWE-RL, SWE-Swiss, SWE-Fixer, etc.). Furthermore, as discussed in Section 2, agent-based training methods are not trainable end-to-end directly from raw GitHub PRs because intermediate steps (planning, reasoning, reflection, root-cause analysis) lack ground-truth labels. To address this, these methods rely on expensive, closed-source LLMs (like Claude-3.5-Sonnet) for curating agent trajectories for training data which is beyond the scope of our work. Notably, our work involves no such dependency on proprietary models and our training dataset is curated solely from the raw PRs written by human developers. Thus, our choice of using a static two-step pipeline significantly reduces the overall computational cost of our work. While agentic scaffolds may offer more flexibility, our choice of a static two-step scaffold is sufficient for answering our core research question: "can open-source LLMs be trained using raw PR data to generate bug reproduction tests given issue descriptions?"

---

### Meta-Review · Area_Chair_x4ZB · 2025-12-30

**Summary:**

Reviewers acknowledged SWE-Tester’s valuable 41K-instance dataset and empirical improvements (up to 10% success rate gain) on open-source LLMs for issue reproduction. Reviewers also raise some concerns regarding the limited methodological novelty (relying on existing static two-step pipelines), overreliance on inference-time sampling/reranking, restricted generalizability to single-test-file edits, and insufficient baselines. While the authors addressed novelty and inference-scaling concerns via rebuttal, the remaining concerns have not been fully addressed during the rebuttal, and it seems that the paper still needs substantial improvements. Therefore I would like to recommend rejection.

**Reviewer Concerns:**

The authors clarified that fine-tuning (not just sampling/reranking) drives performance gains via oracle edit-only experiments, justified the agentless pipeline as practical for open-source data curation, and distinguished issue reproduction from related tasks. They also added partial baseline comparisons and noted multi-file edit support.

However, there still remain several critical gaps: methodological novelty remains limited (adapting existing pipelines), generalizability is constrained to single-test-file scenarios, and comparisons with relevant code agents are still insufficient. The static pipeline’s lack of scalability amid the shift to agentic systems persists as a key limitation.

**Reviewer Scores:**

Only one reviewer mentioned that he/she will retain the score. Regarding the remaining reviewers, they did not share their comments regarding the authors' rebuttal.

Based on the reviewers' original comments and authors' response, I think that the reviewers will likely maintain their original scores as some of their concerns are not well addressed, such as the scalability and flexibility of the proposed method.

---

### Decision · Program_Chairs · 2026-01-26

Reject